# Augmented Global Protein Acetylation Diminishes Cell Growth and Migration of Cholangiocarcinoma Cells

**DOI:** 10.3390/ijms251810170

**Published:** 2024-09-22

**Authors:** Saowaluk Saisomboon, Ryusho Kariya, Panupong Mahalapbutr, Tonkla Insawang, Kanlayanee Sawanyawisuth, Ubon Cha’on, Thanyada Rungrotmongkol, Sopit Wongkham, Sarawut Jitrapakdee, Seiji Okada, Kulthida Vaeteewoottacharn

**Affiliations:** 1Department of Biochemistry, Faculty of Medicine, Khon Kaen University, Khon Kaen 40002, Thailand; saowaluksai@kkumail.com (S.S.); panupma@kku.ac.th (P.M.); skanlaya@kku.ac.th (K.S.); ubocha@kku.ac.th (U.C.); sopit@kku.ac.th (S.W.); 2Division of Hematopoiesis, Joint Research Center for Human Retrovirus Infection and Graduate School Medical Sciences, Kumamoto University, Kumamoto 860-0811, Japan; ryushokariya@gmail.com (R.K.); okadas@kumamoto-u.ac.jp (S.O.); 3Laboratory of Molecular Cellular Biology, School of Pharmaceutical Sciences, Kobe Gakuin University, 1-1-3 Minatojima, Chuo-ku, Kobe 650-8586, Japan; 4Center of Translational Medicine, Faculty of Medicine, Khon Kaen University, Khon Kaen 40002, Thailand; 5Research Instrument Center, Khon Kaen University, Khon Kaen 40002, Thailand; tonkin@kku.ac.th; 6Center of Excellence in Structural and Computational Biology, Department of Biochemistry, Faculty of Science, and Program in Bioinformatics and Computational Biology, Graduated School, Chulalongkorn University, Bangkok 10330, Thailand; thanyada.r@chula.ac.th; 7Department of Biochemistry, Faculty of Science, Mahidol University, Bangkok 10400, Thailand; sarawut.jit@mahidol.ac.th

**Keywords:** acetyl-CoA carboxylase 1 (ACC1), protein hyperacetylation, cholangiocarcinoma (CCA), lysine deacetylase (KDAC) inhibitor, Ak strain transforming (AKT), Snail

## Abstract

We have previously shown that the overexpression of acetyl-CoA carboxylase 1 (ACC1) was associated with the poor prognosis of cholangiocarcinoma (CCA) patients, and suppression of its expression in CCA cell lines deteriorated cell growth. The present study explored the mechanism by which ACC1 inhibition affects global protein acetylation, using genetic knockdown and pharmacological inhibition with an ACC1 inhibitor ND-646 as models. Both ACC1 knockdown and ACC1-inhibitor-treated cells displayed the hyperacetylation of proteins, accompanied by impaired growth and migration. The immunoprecipitation of hyperacetylated proteins using the anti-acetylated lysine antibody, followed by tandem mass spectrometry, identified three potential verification candidates, namely POTE ankyrin domain family member E, peroxisomal biogenesis factor 1, and heat shock protein 90 beta (HSP90B). HSP90 acetylation was the candidate selected for the verification of protein acetylation. To establish the effects of protein hyperacetylation, treatment with suberoylanilide hydroxamic acid (SAHA), a lysine deacetylase inhibitor, was conducted, and this served as an independent model. Decreased tumor growth but increased acetylated protein levels were observed in ACC1-KD xenograft tumors. Hyperacetylated-alleviated cell growth and migration were consistently observed in the SAHA-treated models. The molecular linkage between protein hyperacetylation and the AKT/GSK3β/Snail pathway was demonstrated. This study highlighted the importance of protein acetylation in CCA progression, suggesting that ACC1 and KDAC are potential targets for CCA treatment.

## 1. Introduction

Cholangiocarcinoma (CCA) is a cancer of the bile duct lining epithelia, with an exceptionally high prevalence rate reported in East Asia and Thailand [1,2]. The therapeutic efficacy of the current treatment options is significantly hindered by the substantial heterogeneity of CCAs and delayed diagnosis [1]. Moreover, despite curative surgery intentions, CCA recurrence rates range from 50% to 70% [3]. Therefore, understanding the molecular mechanism of CCA progression might shed some light on the novel treatment for CCA.

Metabolic reprogramming in cancer cells is recognized as one of the critical processes for carcinogenesis and progression [4]. In CCA, the dependence of high glucose on CCA migration and invasion promotion is established [5]. Increased post-translational modification (PTM), particularly O-GlcNAcylation, was observed in glucose-addicted cells [5]. Moreover, a recent report has emphasized the role of the dysregulation of de novo lipogenesis [6], the alternative pathway of glucose-derived products.

Acetyl-CoA carboxylase (ACC) is a biotin-dependent carboxylase enzyme that catalyzes the first-rate limiting step enzyme in de novo lipogenesis. ACC consists of three functional domains, namely, biotin carboxylase (BC), carboxyltransferase (CT), and biotin carboxyl carrier protein (BCCP) domains [7]. In mammals, ACC consists of two isoforms, ACC1 and ACC2. The former is involved in lipogenesis, which is elevated in several human cancers. Furthermore, ACC1 is associated with shorter survival rates and unfavorable clinical outcomes of CCA patients. Targeting ACC1 by gene editing (ACC1-KD) or pharmacological inhibition underscores the requirement of lipogenesis for CCA progression [6,8].

Because acetyl-CoA is a substrate for ACC1 and protein acetylation, the disruption of ACC1 can potentially perturb this post-translational modification [9,10]. The tight regulation of acetylation is crucial in controlling various cellular pathways [11]. Increased protein acetylation leading to reduced cancer cell growth, migration, and angiogenesis was previously demonstrated [10,12,13]. In CCA, lysine deacetylase (KDAC) inhibition suppresses CCA growth and potentiates apoptosis [14,15]. Nevertheless, the mechanisms causing the effects of protein acetylation on CCA growth and migration have never been explored and are the key objectives of the current study.

In the ACC1-KD model, the inhibition of acetyl-CoA carboxylation would result in acetyl-CoA accumulation, potentially increasing protein acetylation. Here, we demonstrated the effects of ACC1 deficiency on protein acetylation in both ACC1 knockdown and ACC1 inhibitor ND-646-treated cells. Three acetylated protein candidates were demonstrated from ACC1-deficient cells. To establish the impact of protein hyperacetylation on CCA functions, a KDAC inhibitor, suberoylanilide hydroxamic acid (SAHA), was applied and used as a supporting model. Our results showed that deprivation of ACC1 induces the hyperacetylation of proteins, consequently inhibiting the growth and migration of CCA cells.

## 2. Results

### 2.1. ND-646 Suppressed CCA Cell Proliferation and Migration

To select an ACC1 candidate inhibitor for in vitro testing, molecular docking stimulation was applied to determine the binding affinities of four well-known ACC inhibitors, including ND-646, soraphen A, CP-640186, and TOFA, with human ACC1. The first two inhibit biotin carboxylase (BC) activity, while the latter two inhibit carboxyltransferase (CT) activity [16]. As shown in Figure 1a, ND-646 had the lowest docking interaction energy (−66.31 kcal/mol), indicating the tightest binding to the BC domain of ACC1. Thus, ND-646 was selected for further studies. Principally, ND-646 prevents ACC phosphorylation, dimerization, and malonyl CoA formation [17].

Clonogenic and Boyden chamber assays were performed to determine the effects of ND-646 on CCA cell proliferation and migration. ND-646 inhibited colony formation (Figure 1b) and cell migration (Figure 1c) in a dose-dependent manner. It should be mentioned that the effects of ND-646 on cell proliferation evaluated by MTT assay at 48 h showed no differences between any of the tested doses. Thus, the discrepancy between the ND-646 growth inhibitory effect observed through clonogenic and MTT assays requires additional exploration beyond the scope of the current work.

### 2.2. Increased Total Protein Acetylation in ACC1 Inhibitory Cells

The suppression of ACCs augmented acetyl-CoA accumulation, which later induced global protein acetylation, was previously reported [9,10]. Thus, the effect of ACC1 inhibition on protein acetylation in CCA cells was investigated. Cells were treated with various concentrations of ND-646 for 24 h, and protein acetylation was assessed by Western blotting, using an anti-acetylated lysine (Ac-K) antibody. The result demonstrated that ND-646 treatment increased global protein acetylation (Figure 2a). Protein hyperacetylation was consistently observed in the CRISPR-Cas9-generated ACC1-deficient CCA cells (ACC1-KD) (Figure 2b). Together, these results confirmed that the modulation of ACC1 by pharmacological inhibition or gene deletion resulted in global protein hyperacetylation in CCA cells.

### 2.3. HSP90B, PEX1, and POTEE Were Acetylated in ACC1-Deficient Cells

To identify the altered acetylated proteins in the ACC1-deficient cells, acetylated proteins were immunoprecipitated under four conditions, including KKU-213A ACC1-KD clone #5 (13AC1-KD#5), KKU-213A ACC1-KD clone #18 (13AC1-KD#18), KKU-213A cells treated with 0.5 µM ND-646 for 24 h, and KKU-213A parental cells, with anti-Ac-K antibody. After pulling-down, SDS-PAGE, and Coomassie blue staining, four observable protein bands were cut (Appendix A) and in-gel tryptic digested, and the peptides were identified by liquid chromatography electrospray ionization tandem mass spectrometry (LC/ESI–MS/MS). All acetylated proteins are demonstrated in Figure 3a and Appendix A. Three candidate proteins, including POTE ankyrin domain family member E (POTEE), peroxisomal biogenesis factor 1 (PEX1), and heat shock protein 90 beta (HSP90B), were commonly detected under ACC1-inhibitory conditions but not in the KKU-213A parental cell line.

The preliminary analysis using the GEPIA tool showed that the expression of HSP90B was significantly higher in the CCA tissues than in the non-cancerous tissues, while that of PEX1 and POTEE was slightly increased (Appendix A). HSP90 was chosen as a candidate for further validation of increased protein acetylation in ACC1-deficient cells. The results confirmed that HSP90 acetylation was increased in both ACC1-deficient conditions (Figure 3b). It should be pointed out that the total HSP90 protein levels were similar under all tested conditions, suggesting that ACC1 modulation affected only the post-translational modification of HSP90 via acetylation.

### 2.4. Inhibition of ACC1 Affected the AKT/GSK3β/Snail Axis

HSP90 is a chaperone protein that stabilizes target protein folding [18]. HSP90 acetylation can bind to several targets and regulate their biological activities, including AKT [19]. AKT has been reported to regulate gastric cancer cell migration by modulating GSK3β activity and Snail expression [20]. The disruption of HSP90–AKT binding results in the dephosphorylation and inactivation of AKT [21]. The crucial role of AKT in CCA migration has previously been demonstrated [22,23]. The above evidence suggests that the attenuation of CCA cell migration in ACC1 deficiency-related protein hyperacetylation may be mediated through the AKT/GSK3β/Snail axis.

To examine the association between ACC1 deprivation and the AKT-GSK3β signaling axis, we measured the expressions of AKT, pAKT, GSK3β, pGSK3β, Snail, and phosphorylated ACC (pACC). The results confirmed the reductions of pACC and ACC1 in ACC1-KD cells, while pACC was diminished in ND-646-treated KKU-055 and KKU-213A. The reduced expressions of pAKT (S473), pGSK3β (S9), and Snail were observed in ACC1-inhibitory cells, particularly in the ND646-treated cells. The levels of total HSP90 remained unaffected (Figure 4a,b). Overall, these results suggested that the depletion of ACC1 attenuated the AKT/GSK3β/Snail pathway in CCA cells.

### 2.5. KDAC Inhibitor SAHA Induced HSP90 Acetylation but Inhibited CCA Cell Proliferation and Migration

To emphasize the relevance of protein hyperacetylation on CCA functions, a well-known KDAC inhibitor, SAHA, was applied [24]. Ac-K levels were determined in SAHA-treated CCA cells, and the results showed that SAHA treatment induced total protein acetylation in a dose-dependent manner (Figure 5a,b). The effects of SAHA treatment on HSP90 acetylation, cell proliferation, and migration in CCA cells were evaluated. The results revealed that acetylated HSP90 levels were increased in SAHA-treated KKU-055 and KKU-213A cells (Figure 5c). SAHA inhibited CCA cell proliferation in a dose- and time-dependent manner (Figure 5d). Moreover, a dose-dependent suppression of the CCA cell migration of SAHA was demonstrated (Figure 5e). These results confirmed that global protein acetylation was consistently observed in both ACC1-inhibitory and KDAC-inhibitory CCA cells, and the common acetylated protein in all tested conditions was HSP90. The growth inhibition and migration retardation were consistently observed under all hyperacetylated protein conditions. It is worth noting that the pattern of global protein acetylation observed in the ND-646-treated cells and the ACC1-KD cells (Figure 2a,b) was distinct from that of the SAHA-treated cells (Figure 5a); hence, additional exploration is required to identify the differential protein acetylation in ACC1-deficient and KDAC-inhibitory CCA cells.

### 2.6. ACC1 Inhibitory Conditions and SAHA Treatment Diminished GSK3β/Snail Signaling in an AKT-Dependent Manner

As demonstrated in Figure 4, ACC1-deprivation conditions promoted HSP90 acetylation and inhibited the AKT/GSK3β/Snail-related pathway. A similar phenomenon was assumed under SAHA-induced hyperacetylation conditions. pAKT (S473), pGSK3β (S9), and Snail were evaluated in SAHA-treated CCA cells. The results showed a similar decrease in pAKT, pGSK3β, and Snail levels under both SAHA-treated cells and ACC1 deficient conditions. To confirm the involvement of AKT in hyperacetylated CCA cells, SC-79 (AKT activator) was used to restore AKT activity. The results demonstrated that GSK3β phosphorylation and Snail expression were increased under all SC-79-treated induced-AKT phosphorylation conditions (Figure 6a,b). These findings emphasized that hyperacetylation status, promoted by ACC1 deficiency or KDAC inhibition, diminished GSK3β/Snail signaling in an AKT-dependent manner.

### 2.7. Decreased Tumor Growth Was Observed in Acquired Hyperacetylated ACC1-Deficient CCA Cells

An in vivo model was employed to emphasize the importance of ACC1 in CCA. Two clones of ACC1-KD (13AC1-KD#5 and 13AC1-KD#18), along with KKU-213A cells, were injected into mice subcutaneously. Tumors were monitored and removed on day 17. The results demonstrated that the tumor weights and volumes of the ACC1-KD-injected groups were markedly smaller than those of the control (Figure 7a–c). In contrast, no significant differences in mice body weight were observed among groups (Figure 7d). It is worth noting that one tumor from the 13AC1-KD#5-injected group and three tumors from the 13AC1-KD#18-injected group were undetectable (open circles) (Figure 7a).

The detections of CK-19 (epithelial marker), Ki-67 (proliferative marker), and ACC1 were performed to emphasize the contribution of ACC1 alleviation to CCA cell growth. The results revealed that the nuclear Ki-67 was markedly reduced in CCA cells with acquired diminished ACC1 expression compared to the control levels (Figure 7e). Ki-67-positive nuclei were detected in 51.7 ± 12.1% and 39.9 ± 10.2% of 13AC1-KD#5 and 13AC1-KD#18 samples compared to the results for KKU-213A (*p* < 0.001, Appendix A). Consistent with the in vitro results, ACC1-KD-xenografted tumor tissues possessed decreased ACC1 but increased Ac-K levels (Figure 7e). It should be noted that increased protein acetylation was observed in both the nuclear and cytoplasmic compartments.

### 2.8. SAHA Treatment Promoted Global Protein Acetylation but Inhibited CCA Growth

To examine the effects of SAHA treatment on CCA growth in vivo, KKU-213A cells were subcutaneously injected into mice. The tumor volume and body weight of the mice were measured daily, and the tumors were removed on day 14. The results showed that SAHA suppressed CCA growth, as demonstrated by reduced tumor size, volume, and weight (Figure 8a–c). The apparent adverse effect of SAHA was not observed when the general appearance and body weights of the mice were monitored (Figure 8d).

The Ac-K level was determined using immunohistochemistry staining to obtain the increased total protein acetylation in SAHA-treated CCA. The results demonstrated that Ac-K was increased, while nuclear Ki-67 and Snail expressions were decreased in the SAHA-treated group. It is relevant to note that the HSP90 levels were comparable between the control and SAHA-treated groups (Figure 8e and Appendix A).

## 3. Discussion

The significance of ACC1-dependent de novo lipogenesis in CCA has been recently reported [6]. The overexpression of ACC1 was associated with shorter survival time in CCA patients. The mechanistic study revealed that the depletion of ACC1 hampered de novo lipogenesis and Snail-mediated cell migration in CCA. In this study, the consequences of acetyl-CoA accumulation on protein acetylation and CCA cell properties were explored, and the importance of protein hyperacetylation in CCA was established by KDAC inhibitor SAHA treatment. The increased protein acetylation was observed in ACC1-inhibitory cells (ND-646 treatment and ACC1-KD). Three common acetylated proteins, namely HSP90, PEX1, and POTEE, were identified under ACC1-inhibitory conditions, and HSP90 acetylation was confirmed. The crucial role of AKT in hyperacetylation-controlling GSK3β/Snail signaling was established using an AKT activator SC-79 and KDAC inhibitor SAHA. Elevated protein acetylation but decreased CCA growth was consistently observed in BRJ-bearing ACC1-KD cells and under SAHA-treated conditions. This study is the first to suggest hyperacetylation promotion via ACC1 or KDAC inhibition for CCA treatment.

The pharmacological inhibition of ACC1 by ND-646-suppressed CCA cell growth and migration was consistent with the results of a previous study in non-small cell lung cancer (NSCLC) [17]. Genetic deletion and pharmacological inhibition of ACC1 attenuated NSCLC cell growth in vitro and in vivo [17]. Protein acetylation is a vital process in cancer progression [11]. Hyperacetylation-promoted cancer-migrated properties observed in this study are consistent with the results of a previous report in breast cancer [10]. Increased smad2 acetylation primed EMT, followed by metastasis, was demonstrated in 4T1 cells. However, enhanced protein acetylation by ACC1 inhibition and acetyl-CoA accumulation leads to the activations of inflammation- and anti-apoptotic-related genes in prostate (PC3) and ovarian (OVCAR3) cancer cells [9], suggesting the outcomes of acetyl-CoA accumulation appear to be cell type-dependent. Additional experiments to identify predictive factors are warranted for further investigation.

Among three identified acetylated proteins, increased HSP90 acetylation was confirmed under all ACC1-inhibitory conditions. Principally, HSP90 acts as a chaperon protein that modulates the target protein’s stabilities and functions [25]. HSP90 overexpression was observed in various types of cancers. Inhibition of HSP90 through disruption of the ATP binding [26], interaction with the co-chaperone [27], and acetylation [28] has been shown to suppress cancer progression. Acetylated HSP90 facilitates the target protein degradation via the proteasome system, as reported in breast cancer [28]. The association between HSP90 acetylation and reduced CCA cell growth and migration was consistently observed in the present study, suggesting that aberrant HSP90 acetylation may contribute to the attenuation of CCA oncogenic properties.

AKT is a well-known HSP90 target [21], and it plays an important role in CCA cell survival, migration, and invasion [22,23]. Decreased pAKT was noted in ACC1-inhibitory cells, and reduced GSK3β phosphorylation was observed in CCA cells with decreased pAKT. Reduced levels of Ki-67 and Snail confirmed the alterations of CCA cell growth and migration in ACC1-KD cells in vitro and in vivo. AKT-regulated GSK3β activity was associated with growth in breast cancer [29] and migration in gastric cancer cells [20]. The current study suggested that diminishing ACC1 promoted HSP90 acetylation and impeded CCA cell growth and migration. These effects might be due to the AKT/GSK3β pathway.

To confirm the relevance of protein hyperacetylation in CCA, the current study utilized the KDAC inhibitor SAHA and demonstrated its effects under in vitro and in vivo conditions. Suppressions of CCA cell proliferation and migration were detected under SAHA-treated conditions. Consistent observations of increased global protein acetylation, HSP90 acetylation, and the suppression of the AKT/GSK3β/Snail axis were revealed in SAHA-treated cells. The functional importance of AKT in protein acetylation-mediated GSK3β/Snail activation was established using AKT activator SC-79 treatment. The suppression of KDAC, particularly KDAC6, affected global protein acetylation, and HSP90 acetylation was previously reported in other cancers [19,30,31]. The hyperacetylation of HSP90 reduces cancer cell growth, migration, and angiogenesis, mainly through the reduction of the protein–protein interaction and the induction of protein degradation. Potential uses of KDAC inhibitors for CCA treatment were previously demonstrated [14,15]. KDAC inhibition using KDAC3-specific or pan inhibitors affected CCA growth suppression and apoptosis induction. Ultimately, increased protein acetylation-suppressed CCA migration was initially demonstrated in the current study. The emerging impact of protein hyperacetylation on CCA cell properties, including growth and migration, were established herein; hence, this method may prove to be an alternative targeted therapy for CCA, particularly in cases with increased ACC1 or KDAC. Moreover, the potential use of HSP90-targeted therapy is suggested.

## 4. Materials and Methods

### 4.1. Cell Lines and Cell Culture

The CCA cell lines KKU-055 (RRID: CVCL_M258) and KKU-213A (RRID:CVCL_M261) [32] were obtained from the Japanese Collection of Research Bioresources (JCRB), Japan. The ACC1-deficient (ACC1-KD) cell lines 13AC1-KD#5 (#5) and 13AC1-KD#18 (#18) were established from a previous study [6]. The cells were maintained in Dulbecco’s modified Eagle’s medium (DMEM) (Wako Pure Chemicals, Osaka, Japan), containing 10% fetal bovine serum (FBS) (Biowest, Riverside, MO, USA) and 1x antibiotic-antimycotic, in a humidified incubator at 37 °C with 5% CO_2_.

### 4.2. Antibodies and Reagents

The antibodies were obtained from several sources, as follows: anti-ACC1 (#21923-1-AP) was procured from Proteintech Group Inc. (Rosemont, IL, USA); anti-phosphoACC (Ser79; #3661), anti-acetylated-lysine (Ac-K) (#9441), anti-AKT (C67E7; #4691), anti-phosphoAKT (Ser473; D9E; #4060), anti-GSK3β (27C10; #9315), anti-phosphoGSK3β (Ser9; 5B3; #9323), anti-HSP90 (#4874), anti-Snail (C15D3; #3879), horseradish peroxidase (HRP)-linked goat anti-rabbit IgG (#7074), and HRP-linked horse anti-mouse IgG (#7076) were purchased from Cell Signaling Technology, Inc., (Danvers, MA, USA); anti-actin (C-2; #8432) was obtained from Santa Cruz Biotechnology (Santa Cruz, CA, USA); anti-KRT19 (#HPA002465) was procured from Sigma Aldrich (St. Louis, MO, USA); biotinylated goat anti-mouse IgG antibody (#BA-9200) and biotinylated goat anti-rabbit IgG antibody (#BA-1000) were purchased from Vector laboratories Inc., (Newark, CA, USA).

ND-646 (#HY-101842) was purchased from MedChemExpress (Monmouth Junction, NJ, USA). *N*-hydroxy-*N’*-phenyloctanediamide or suberoylanilide hydroxamic acid (SAHA) (#H1388) was obtained from the Tokyo Chemical Industry (Tokyo, Japan). SC-79, Akt Activator II (#123871), was obtained from EMD Millipore (Billerica, MA, USA).

### 4.3. Molecular Docking Analysis

The homology models of the human ACC1 biotin carboxylase (BC) and carboxyl transferase (CT) domains were constructed by the SWISS-MODEL web server, using the crystal structures of the human ACC2 BC domain, with ND-646 (PDB entry 5KKN) [33], and the human ACC2 CT domain, with CP-640186 (PDB entry 3FF6) [34] as the template. The structures of ND-646, soraphen A, CP-640186, and TOFA were downloaded from the PubChem database [35] and subjected to the protonation state evaluation at pH 7.0 using the MarvinSketch program, Marvin 23.5.0, 2023, ChemAxon (http://www.chemaxon.com). Molecular docking simulation was performed using the CDOCKER module [36], implemented in the Discovery Studio 2.5 program, with a docking sphere radius of 15 Å.

### 4.4. Clonogenic Assay

KKU-055 and KKU-213A were seeded overnight into 12-well plates (200 cells/well). Next, the cells were treated with 0, 0.1, 0.5, and 1.0 µM ND-646 for 7 days. ND-646-containing media was replaced every 3 days. Then, the cells were fixed with 4% paraformaldehyde and stained with 0.1% crystal violet. The cell number was calculated from the OD at 595 nm by dissolving the crystal violet with 1% acetic acid.

### 4.5. Cell Migration Assay

The Boyden chamber, containing an 8.0 μm pore insert (Corning, NY, USA), was used to determine cell migration. KKU-055 (5 × 10^4^ cells) and KKU-213A (3 × 10^4^ cells), mounted on Transwell inserts, were treated with various concentrations of ND-646 or SAHA for 24 h. Crystal violet was used as a visualization aid. The dye was dissolved with 1% acetic acid, and absorbance was measured at 595 nm.

### 4.6. MTT Assay

A total of 3000 CCA cells were plated into a 96-well plate overnight before various concentrations of SAHA (0–5 µM) were applied to the cells for 24, 48, and 72 h. At the indicated time points, the 3-(4,5-dimethylthiazol-2-yl)-2,5-diphenyltetrazolium bromide (MTT) (Invitrogen; Carlsbad, CA, USA) was added to yield a final concentration of 0.5 mg/mL. Subsequently, acid isopropanol (0.04 N HCl in isopropanol) was added to dissolve the formazan complex, and the absorbance at 570 nm was measured.

### 4.7. Protein Extraction and Western Blot Analysis

The cells were lysed in lysis buffer (50 mM Tris-HCl, pH 7.4, 150 mM NaCl, 1% NP-40, 1 mM NaF, 1 mM Na_3_VO_4_) containing a protease inhibitor cocktail (Nacalai Tesque, Kyoto, Japan). The protein concentrations were quantified by bicinchoninic acid (BCA) protein assay (Thermo Science, Rockford, IL, USA). The proteins were subject to SDS-PAGE under reducing conditions and blotted onto a PVDF membrane (GE Healthcare Japan, Tokyo, Japan). After blocking the non-specific signals, the selected primary antibodies and horseradish peroxidase-conjugated secondary antibodies were added. Proteins were detected using Chemi-Lumi One Super reagents (Nacalai Tesque, Kyoto, Japan) and visualized using an ImageQuant LAS800 system (GE Healthcare Japan, Tokyo, Japan). β-actin served as the internal control. The band intensities were quantified by ImageJ software version 1.53k [37].

### 4.8. Immunoprecipitation

The cells were lysed using lysis buffer containing 25 mM HEPES, 10 mM Na_4_P_2_O_7_·10 H_2_O, 100 mM NaF, 5 mM EDTA, 2 mM NaVO_4_, and 1% Triton X-100. A total of 400 micrograms of the proteins were incubated overnight with 2 µg anti-Ac-K or anti-HSP90 at 4 °C on the vertical rotator. Protein-G Sepharose beads 4 (#71708300AI, GE Healthcare Japan, Tokyo, Japan) were added and rotated at 4 °C for an additional 2 h. The mixtures were centrifuged at 5000 rpm for 5 min and washed 5 times with lysis buffer. After washing, the bounded proteins were dissolved in SDS sample loading buffer and subjected to SDS-PAGE and Western blot analysis.

### 4.9. Identification of Acetylated Peptides Using In-Gel Digestion, Liquid Chromatography–Tandem Mass Spectrometry Analysis

CCA cells were plated using four conditions, i.e., KKU-213A (213A), two ACC1-KD clones: 13AC1-KD#5 (#5) and 13AC1-KD#18 (#18), and 0.5 µM ND-646 treatment for 24 h. Then, cell lysates were prepared, and acetylated proteins were enriched by immunoprecipitation using the Ac-K antibody. The proteins were separated using SDS-PAGE and visualized via Coomassie blue staining. The candidate protein bands were cut and digested by trypsin, following the In-gel tryptic digestion protocol provided by the manufacturer (Pierce™ Trypsin Protease, MS Grade, Thermo Fischer Scientific, Rockford, IL, USA). The tryptic-digested peptides were analyzed using a nano-liquid chromatography system (EASY-nLC II, Thermo Fischer Scientific), coupled to a quadrupole time-of-flight mass spectrometer (micrOTOF-Q II) equipped with nano-electrospray ionization (ESI) (Bruker Daltonik GmbH, Bremen, Germany). LC/ESI–MS/MS spectra were analyzed using Bruker Compass Data Analysis, v.4.0. Compound lists were exported as Mascot generic files (MGF) for a further Mascot program search [38]. Protein identification was performed by searching against the *Homo sapiens* (human) proteins from the SwissProt protein databases [39] and the Mascot MS/MS Ion search engine (www.matrixscience.com) (Matrix Science, London, UK), with the initial searching parameters “enzyme” and “trypsin”, which allowed up to three missed cleavages; carbamidomethylation as a fixed modification; oxidation (HW) and oxidation (M) as variable modifications; a peptide mass tolerance of 0.5 Da and a fragment mass tolerance of 0.6 Da; a peptide charge state of +1, +2, +3; instrument type (ESI-QUAD-TOF); and report top (Auto). The instrument was under the service of the Khon Kaen University Research Instrument Center, Thailand.

### 4.10. Animal Experiments

The growth of the ACC1-KD model was assessed as follows: 15 mice were divided into three groups (5 mice/group), including those transplanted with KKU-213A (213A), 13AC1-KD#5 (#5), and 13AC1-KD#18 (#18). A total of 100,000 CCA cells were subcutaneously injected into both flanks of 6–8-week-old male Balb/c Rag-2/Jak3 double-deficient (BRJ) mice [40]. Body weight and tumor volume were monitored every 3 days. On day 17, the mice were euthanized by cervical dislocation.

To determine the effects of SAHA treatment on CCA growth, KKU-213A (1 × 10^5^ cells) was subcutaneously injected into both flanks of 11 male BRJ mice. The mice were randomly divided into two groups (6 mice/control group and 5 mice/treatment group). On day 3 after CCA injection, SAHA was intraperitoneally injected into the treatment group at a final concentration of 100 mg/kg, 5 days/week, for 2 weeks. DMSO was administered to the control group. Body weight and tumor volume were monitored daily. The mice were euthanized on day 14.

The BRJ mice were kindly provided by Prof. Seiji Okada (Kumamoto University, Japan). The mice were bred, housed, and monitored in the Center for Animal Resources and Development, Kumamoto University, Japan, according to the institutional guidelines, which complied with ARRIVE guidelines 2.0 [41]. In brief, the mice were housed in a controlled environment (22 ± 2 °C, 50 ± 10% humidity, and 12-h light/dark cycle). Food and water were provided ad libitum. The health and behavior of the mice were monitored every 3 days to evaluate humane endpoints. All protocols were approved by the Institutional Animal Care and Use Committee of Kumamoto University, Japan (A2021-053). After euthanasia, the tumors were collected, fixed in 4% paraformaldehyde, and embedded in paraffin for immunohistochemistry staining.

### 4.11. Immunohistochemistry Staining

Paraffin-embedded tissues were prepared using the standard protocol [42], and 5-µm thickness paraffin sections were employed. The expressions of CK-19, Ki-67, ACC1, Ac-K, HSP90, and Snail were evaluated by immunohistochemistry staining. The signals were amplified using the Vectastain elite ABC standard kit (#PK-6100, Vector laboratories Inc., Newark, CA, USA) and 3,3′-diaminobenzidine tetrahydrochloride (DAB) reagent (#425011, Nichirei Bioscience Inc., Tokyo, Japan). Images were captured using the Olympus BX53 microscope (Olympus, Tokyo, Japan).

The Ki-67-positive nuclei were counted using ImageJ software (*n* = 5/group, 20× objective lens). The percentage of Ki-67-positive nuclei was calculated as follows: %Ki-67-positive nuclei = (Ki-67-positive nuclei/total nuclei) × 100. The Ki-67-positive nuclei of the control group were adjusted to 100%.

### 4.12. Gene Expression Profiling from Databases

The expressions of *HSP90*, *PEX1*, and *POTEE* of CCA (CHOL) and comparative noncancerous tissues were obtained from The Cancer Genome Atlas (TCGA) database via Gene Expression Profiling Interactive Analysis (GEPIA, http://gepia.cancer-pku.cn/) [43]. A set of 36 CCA and nine comparative noncancerous tissues were included.

### 4.13. Statistical Analysis and Illustration Creation

SPSS version 23.0 (SPSS Inc., Chicago, IL, USA) and GraphPad Prism^®^8.0.2 Software (GraphPad Software Inc., San Diego, CA, USA) were used for statistical analysis. The Student’s *t*-test was used to compare the differences between treatment and control groups. A *p*-value < 0.05 was considered statistically significant.

The proposed mechanism was created with BioRender (https://biorender.com/).

## 5. Conclusions

This study highlighted the importance of protein hyperacetylation in CCA diminution. Weakening ACC1 or KDAC would induce global protein acetylation, including that of HSP90, and partially suppress CCA cell growth and migration via the AKT/GSK3β axis. The proposed mechanism is demonstrated in Figure 9.

## Figures and Tables

**Figure 1 ijms-25-10170-f001:**
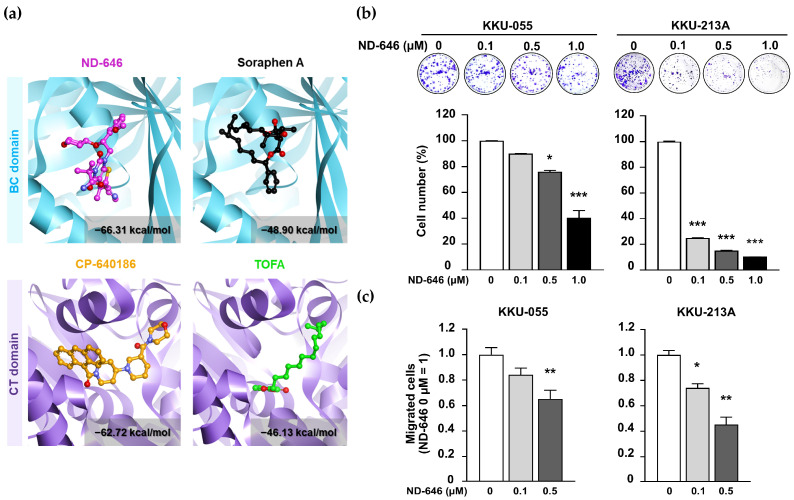
ND-646 inhibited CCA cell proliferation and migration. (**a**) Structures of ND-646 and soraphen A binding with the ACC1 biotin carboxylase (BC) domain (**top**) and CP-640186 and TOFA binding with the ACC1 carboxyltransferase (CT) domain (**bottom**). The interaction energy is demonstrated with a gray box. KKU-055 and KKU-213A were treated with 0, 0.1, 0.5, and 1.0 µM ND-646. (**b**) Cell growth was determined by clonogenic assay on day 7; (**c**) cell migration was evaluated using a Boyden chamber assay at 24 h. The statistical comparison between untreated (0 µM) and ND-646-treated cells is shown as: * *p* < 0.05, ** *p* < 0.01, and *** *p* < 0.001.

**Figure 2 ijms-25-10170-f002:**
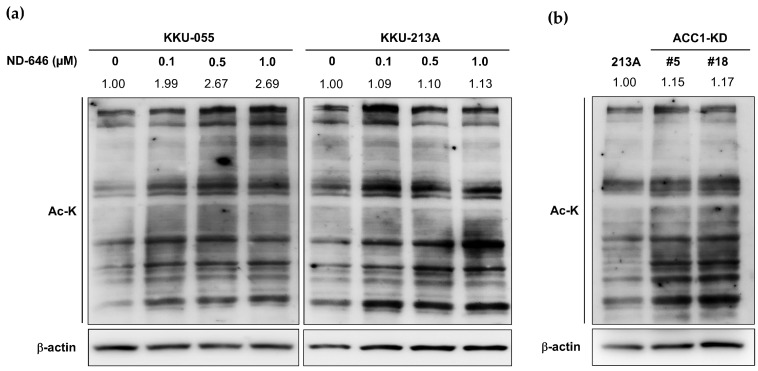
ND-646 and ACC1-KD increased total protein acetylation in CCA cells. (**a**) KKU-055 and KKU-213A were treated with 0, 0.1, 0.5, and 1.0 µM ND-646 for 24 h. (**b**) ACC1-KD and KKU-213A parental cells were cultured for 48 h. Total protein acetylation (Ac-K) was assessed by Western blotting. The intensity of each band was normalized with that of β-actin and is shown as the relative value of the control, which was arbitrarily set as 1 (ND-646 0 μM or 213A = 1), and indicated by the number on the top of each band. ACC1; acetyl-CoA carboxylase 1; Ac-K; acetylated-lysine.

**Figure 3 ijms-25-10170-f003:**
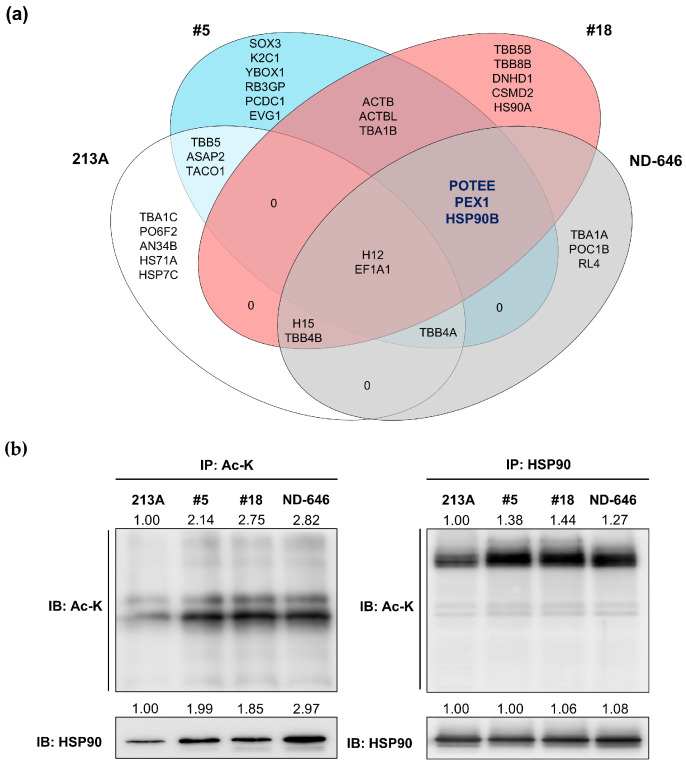
HSP90 hyperacetylation was observed in ACC1 inhibitory cells. Acetylated proteins were compared between KKU-213A (213A) and three ACC1 inhibitory conditions, including two ACC1-KDs, i.e., 13AC1-KD#5 (#5), 13AC1-KD#18 (#18), and 0.5 µM ND-646-treated cells. The acetylated proteins were immunoprecipitated with anti-Ac-K antibody, separated by SDS-PAGE, and in-gel tryptic digested, and the peptides were analyzed by LC/ESI–MS/MS. (**a**) The Venn diagram represents acetylated peptides identified under four conditions. Common peptides detected under all ACC1 inhibitory conditions are presented in blue characters. (**b**) HSP90 hyperacetylation was confirmed under ACC1 inhibitory conditions. Cell lysates were immunoprecipitated using anti-Ac-K and immunodetected by anti-HSP90 (**left**) and vice versa (**right**). The band intensity of HSP90 under each condition was relative to that of control (KKU-213A), which was arbitrarily set as 1, and is shown by the number on the top of each band. Ac-K; acetylated-lysine; HSP90; heat shock protein 90; IB; immunoblotting; IP; immunoprecipitation.

**Figure 4 ijms-25-10170-f004:**
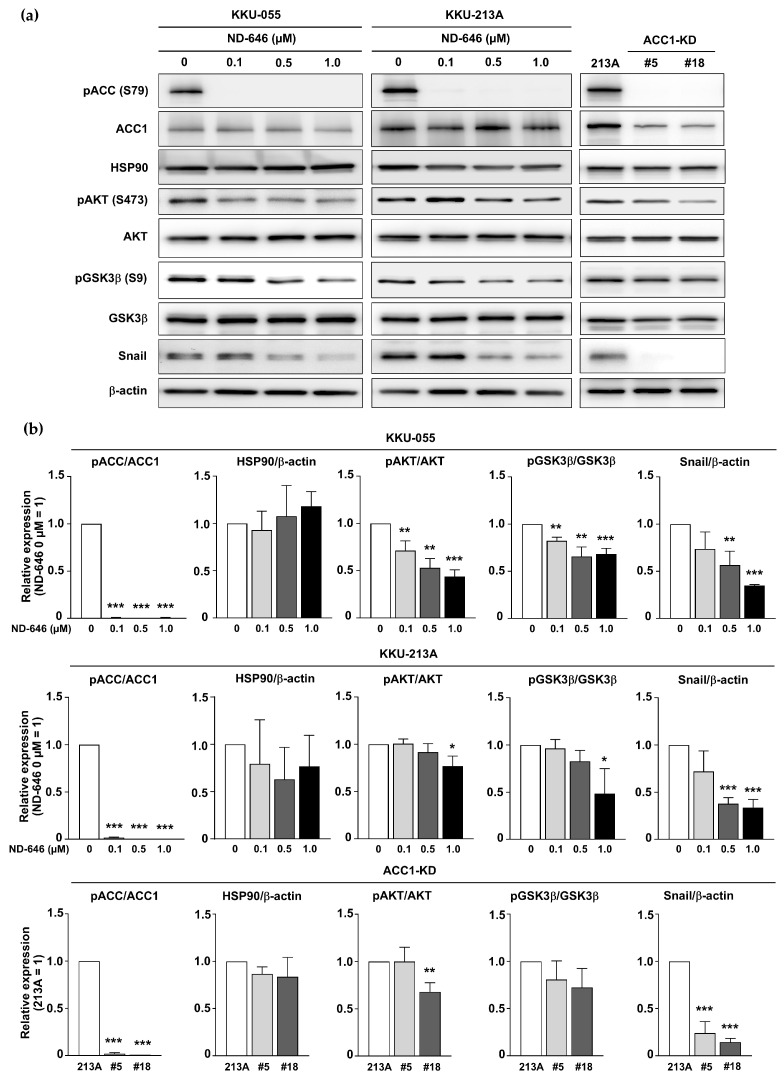
ACC1 inhibitory cells acquired reduced AKT/GSK3β phosphorylation and Snail expression. (**a**) The alterations of pAKT, pGSK3β, and Snail in ND-646-treated KKU-055, KKU-213A, and two ACC1-KD clones—13AC1-KD#5 (#5) and 13AC1-KD#18 (#18)—were determined. β-actin serves as a protein-loading control. (**b**) The ratios of pACC/ACC1, HSP90/β-actin, pAKT/AKT, pGSK3β/GSK3β, and Snail/β-actin were compared between untreated and ND-646-treated cells and between parental cell line (213A) and ACC1-KD cells. * *p* < 0.05, ** *p* < 0.01, and *** *p* < 0.001. ACC1, acetyl-CoA carboxylase 1; AKT, Ak strain transforming; GSK3β, glycogen synthase kinase-3 beta; HSP90, heat shock protein 90; pACC (S79), phosphorylated acetyl-CoA carboxylase at serine 79; pAKT (S473), phosphorylated AKT at serine 473; pGSK3β (S9), phosphorylated glycogen synthase kinase-3 beta at serine 9.

**Figure 5 ijms-25-10170-f005:**
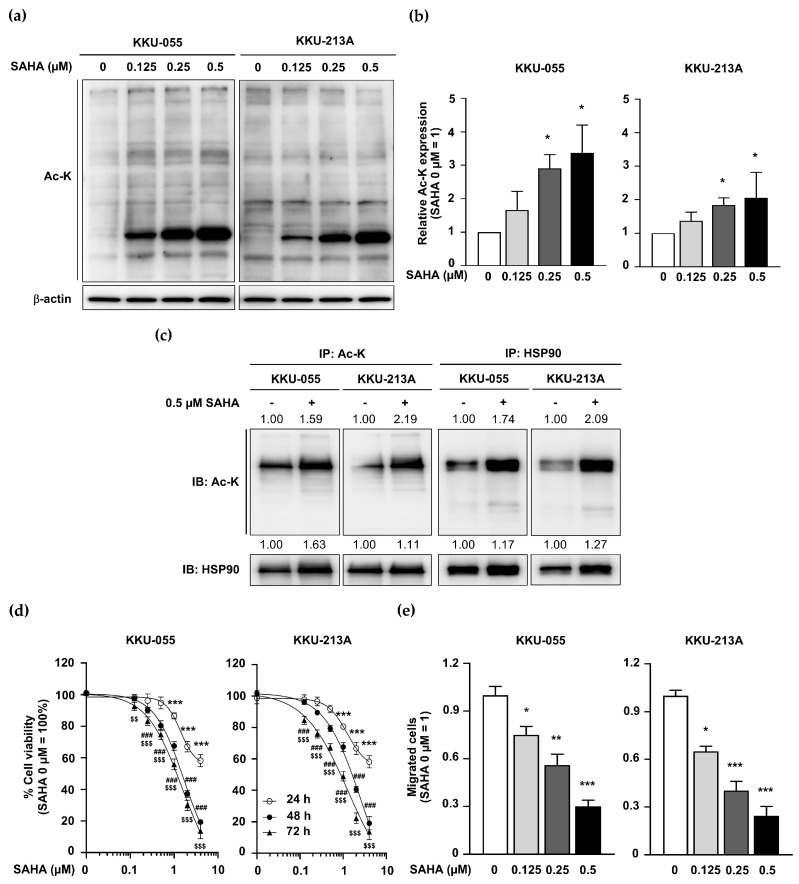
SAHA promoted protein hyperacetylation, particularly HSP90, but inhibited CCA cell growth and migration. (**a**) Total acetylated proteins (Ac-K) were determined by Western blotting in 0–0.5 μM SAHA-treated KKU-055 and KKU-213A. (**b**) Intensities of total Ac-K were quantitated by ImageJ. Relative Ac-K levels are presented in the graph. The total acetylated protein level in the untreated cells is used for normalization. (**c**) HSP90 hyperacetylation was confirmed in SAHA-treated KKU-055 and KKU-213A. Cell lysates were immunoprecipitated using anti-Ac-K and immunoblotted by anti-HSP90 (**left**) and vice versa (**right**). Each band intensity was compared to that of the control (untreated condition = 1), as indicated by the number on the top of each band. (**d**) Cell numbers were determined by MTT assay in untreated and SAHA-treated KKU-055 and KKU-213A at 24, 48, and 72 h. The percentage of cell viability of untreated cells is adjusted to 100%. The statistical significance between untreated and SAHA-treated samples at each time point are as follows: *** *p* < 0.001 at 24 h; ^###^ *p* < 0.001 at 48 h; ^$$^ *p* < 0.01 and ^$$$^ *p* < 0.001 at 72 h. (**e**) Cell migration was evaluated by Boyden chamber assay at 24 h. The statistical comparison between untreated and SAHA-treated conditions is as follows: * *p* < 0.05, ** *p* < 0.01, and *** *p* < 0.001. Ac-K, acetylated-lysine; HSP90, heat shock protein 90; IB, immunoblotting; IP, immunoprecipitation.

**Figure 6 ijms-25-10170-f006:**
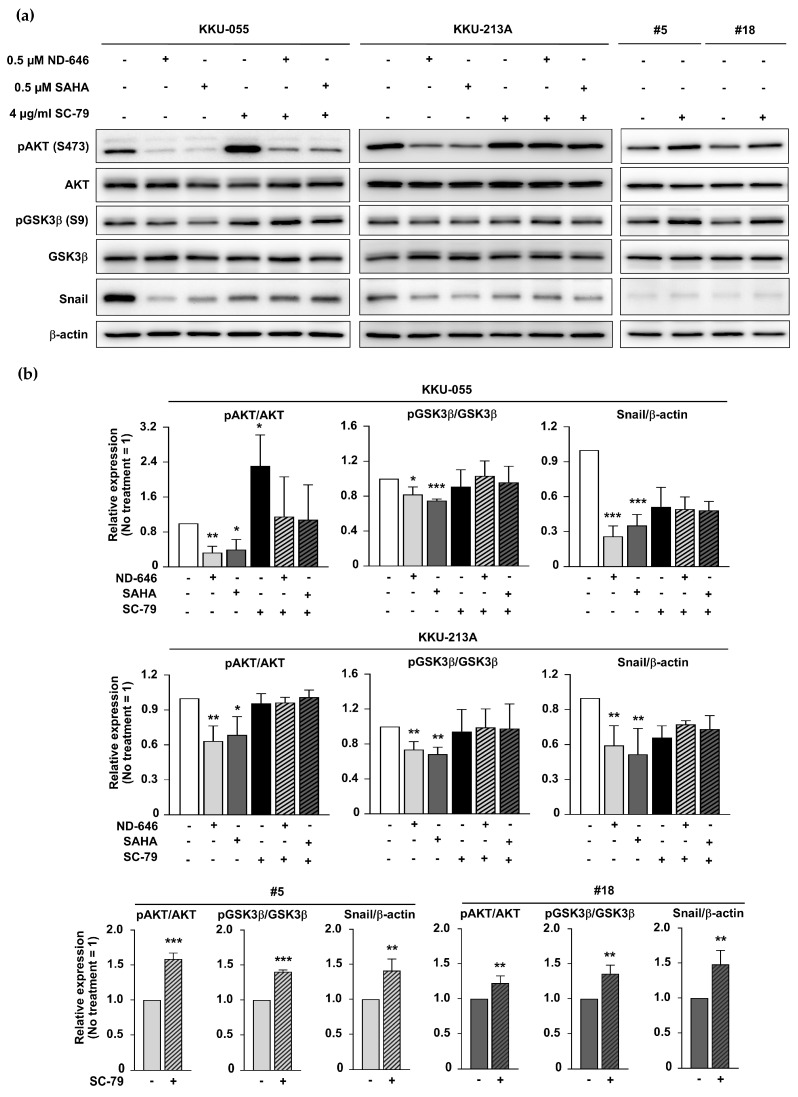
GSK3β/Snail axis in ACC1 inhibitory cells (ND-646-treated and ACC1-KD cells), and KDAC-suppressed (SAHA-treated) cells were regulated by AKT activation. (**a**) The expression of pAKT, pGSK3β, and Snail in ACC1 inhibitory and KDAC inhibitory conditions was detected at 24 h treatment. SC-79 treatment was applied to activate AKT phosphorylation. β-actin serves as a protein-loading control. (**b**) The ratios of pAKT/AKT, pGSK3β/GSK3β, and Snail/β-actin were compared between untreated and treated cells (untreated condition = 1). * *p* < 0.05, ** *p* < 0.01, and *** *p* < 0.001. #5 and #18, 213A-ACC1-KD clones #5 and #18; AKT, Ak strain transforming; GSK3β, glycogen synthase kinase-3 beta; pAKT (S473), phosphorylated AKT at serine 473; pGSK3β (S9), phosphorylated glycogen synthase kinase-3 beta (GSK3β) at serine 9.

**Figure 7 ijms-25-10170-f007:**
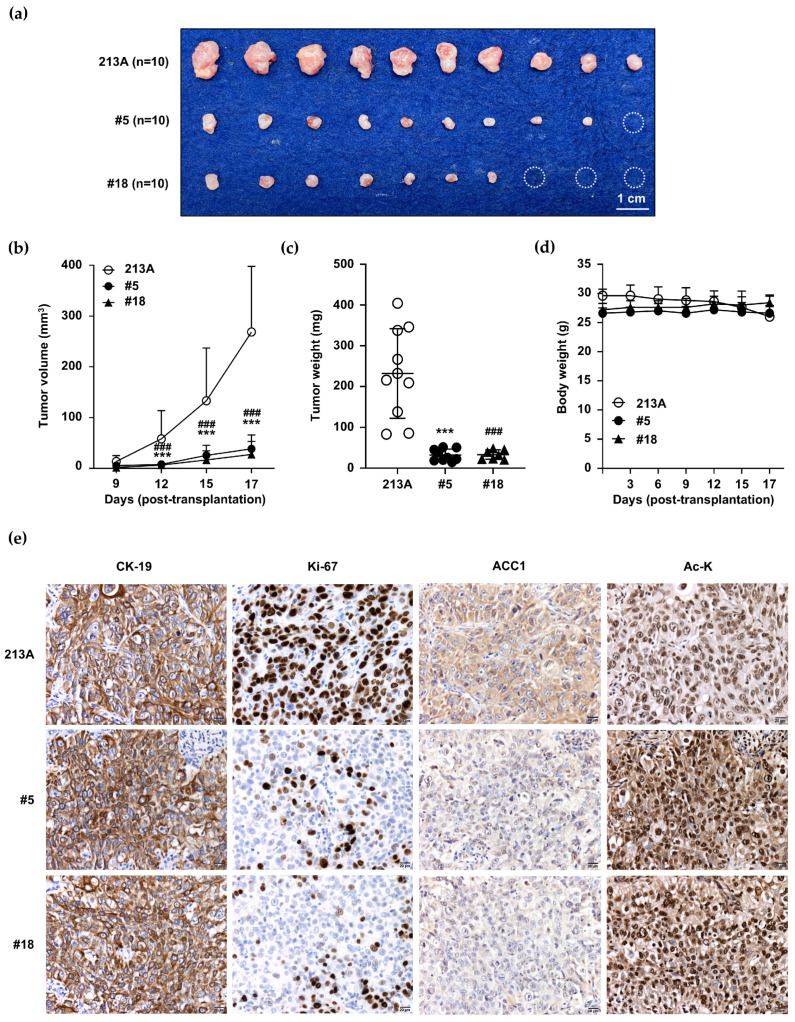
Inhibition of ACC1 expression suppressed CCA cell growth and induced protein acetylation in the BRJ xenograft model. KKU-213A (213A) and two ACC1-deficient (ACC1-KD) clones, 213A-ACC1-KD clone#5 (13AC1-KD#5) and #18 (13AC1-KD#18), were subcutaneously injected into the flanks of mice. On day 17, (**a**) tumors were removed and compared. (**b**) Tumor volumes and (**d**) mice body weights were measured every 3 days and are shown as the means ± SD. (**c**) Tumor weights were measured on day 17 (*n* = 10/group). (**e**) The expressions of Ac-K, ACC1, CK-19, and Ki-67 were compared between 213A and the ACC1-KD clones. The dashed circles denoted the missing tumors. The statistical significance of each ACC1-KD clone compared with 213A is shown as follows: *** *p* < 0.001 for ACC1-KD clone#5, and ^###^ *p* < 0.001 for ACC1-KD clone#18. Ac-K, acetylated-lysine; ACC1, acetyl-CoA carboxylase 1; CK-19, cytokeratin 19; Ki-67, Ki-67 antigen.

**Figure 8 ijms-25-10170-f008:**
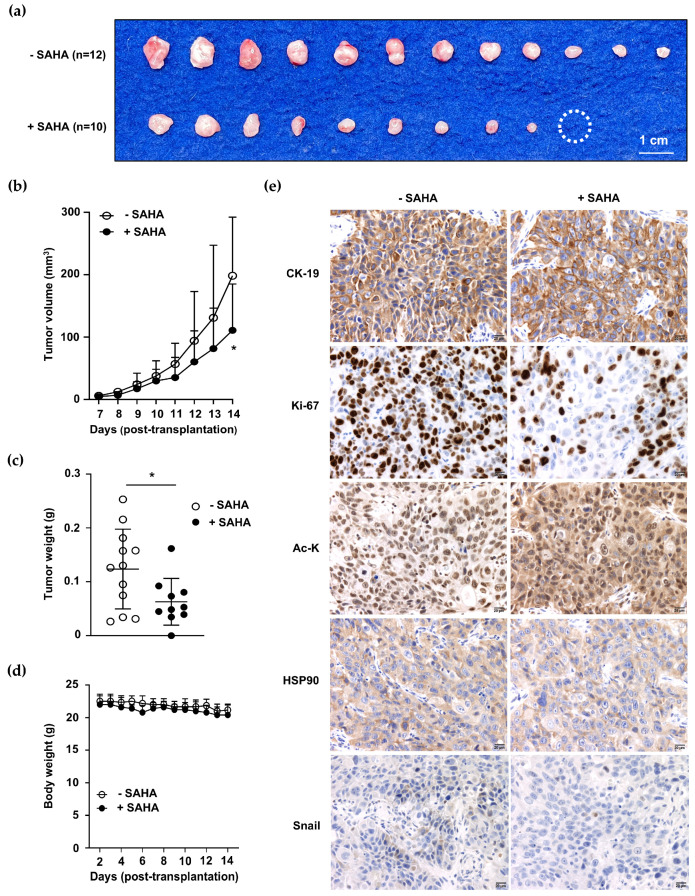
SAHA inhibits cell growth and increases acetylated protein in the BRJ xenografted mouse model. KKU-213A cells were subcutaneously injected into the flanks of mice. SAHA was treated intraperitoneally at 100 mg/kg, 5 days/week, for 2 weeks (*n* = 10). The mice in the control group were treated with an equal volume of DMSO (*n* = 12). (**a**) CCA tumor tissues were obtained from control and SAHA-treated groups after 2 weeks of SAHA treatment. (**b**) Tumor volumes were measured daily. (**c**) Tumor weights were compared. (**d**) Mice body weights were evaluated daily to monitor the general wellbeing of the mice. (**e**) The expressions of CK-19, Ki-67, Ac-K, HSP90, and Snail were compared between untreated (−SAHA) and SAHA-treated (+SAHA) groups. The dashed circles noted missing tumors; * *p* < 0.05. Ac-K, acetylated-lysine; CK-19, cytokeratin 19; HSP90, heat shock protein 90; Ki-67, Ki-67 antigen.

**Figure 9 ijms-25-10170-f009:**
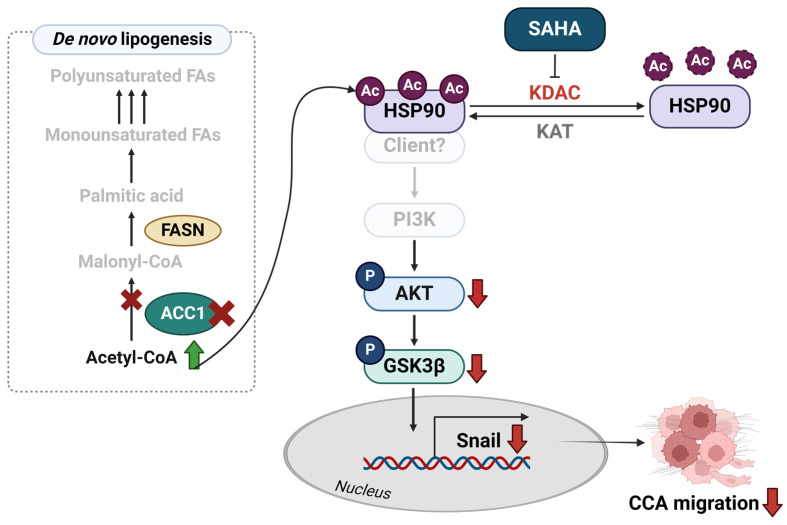
The proposed mechanism suggests that increased global protein acetylation suppresses the migration of CCA cells, partially through an AKT/GSK3β/Snail-related pathway. Protein hyperacetylation is observed in ACC1-deficient or SAHA-treated CCA cells. Increased HSP90 acetylation is demonstrated, accompanied by the suppression of CCA growth and migration. The contribution of AKT/GSK3β to Snail expression and cell migration is shown in the hyperacetylated CCA cells. Ac, acetylation; ACC1, acetyl-CoA carboxylase 1; AKT, Ak strain transforming; FAs, fatty acids; FASN, fatty acid synthase; GSK3β, glycogen synthase kinase-3 beta; HSP90, heat shock protein 90; KAT, lysine acetyltransferase; KDAC, lysine deacetylase; P, phosphorylation; PI3K, phosphatidylinositol 3-kinase.

## Data Availability

The data presented in this study are available on request from the corresponding author.

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
