# Peer review of "Augmented Global Protein Acetylation Diminishes Cell Growth and Migration of Cholangiocarcinoma Cells"

_ijms, 2024, doi:10.3390/ijms251810170_

Round 1
Reviewer 1 Report
Comments and Suggestions for Authors
In the research article by Saisomboon et al., titled “Acetyl-CoA carboxylase 1-mediated protein acetylation and AKT/GSK3β/snail-dependent cell migration in cholangiocarcinoma,” the authors emphasize the role of protein acetylation in cholangiocarcinoma progression, suggesting that ACC1 and KDAC could be potential therapeutic targets for treating this type of cancer. The study's design and manuscript drafting are commendable. The abstract is well-written and conclusive, and the introduction and results sections are detailed and well-structured. The summary figure effectively captures the key findings. However, the manuscript would benefit from language improvements, specifically addressing some grammatical errors. The final line of the abstract, "(200/200)," needs correction. Figure 9 should be elaborated further for clarity and ease of understanding by the readers. Additionally, in the molecular docking studies, it is essential to include the grid dimensions where the compounds were docked.
Comments on the Quality of English LanguageThe manuscript would benefit from language improvements, specifically addressing some grammatical errors.
Author Response
Response to reviewers
Reviewer#1
Comments and Suggestions for Authors
In the research article by Saisomboon et al., titled “Acetyl-CoA carboxylase 1-mediated protein acetylation and AKT/GSK3β/snail-dependent cell migration in cholangiocarcinoma,” the authors emphasize the role of protein acetylation in cholangiocarcinoma progression, suggesting that ACC1 and KDAC could be potential therapeutic targets for treating this type of cancer. The study's design and manuscript drafting are commendable. The abstract is well-written and conclusive, and the introduction and results sections are detailed and well-structured. The summary figure effectively captures the key findings. However, the manuscript would benefit from language improvements, specifically addressing some grammatical errors.
- The final line of the abstract, "(200/200)," needs correction.
Authors: Thank you very much for the suggestion. The abstract word count “(200/200)” was removed.
- Figure 9 should be elaborated further for clarity and ease of understanding by the readers.
Authors: Thank you very much for the recommendation. Figure 9 legend has been modified with additional explanation.
- Additionally, in the molecular docking studies, it is essential to include the grid dimensions where the compounds were docked.
Authors: Thank you very much. Your insightful recommendation has significantly improved the clarity of our work. The grid dimension was included in Materials and Methods “4.3 Molecular Docking Analysis”.
Comments on the Quality of English Language
The manuscript would benefit from language improvements, specifically addressing some grammatical errors.
Authors: Thank you very much for the helpful recommendation. The English language in the current manuscript has been thoroughly reviewed and improved.
Reviewer 2 Report
Comments and Suggestions for Authors
The manuscript titled,” Acetyl-CoA carboxylase 1-mediated protein acetylation and AKT/GSK3β/snail-dependent cell migration in cholangiocarcinoma” has focused on a very interesting topic. My suggestions are as follows:
1. The manuscript needs to be edited by a native English-speaking editor.
2. For all IP experiment, the IP quality in terms of the antibody used for IP probing should be used for western. Besides this author should use IgG as a control antibody.
3. Table 1 can be split on the basis of gender.
4. Authors should also highlight on the strategies followed internationally and the occurrence trend of these type of infertility
5. All IHC data should be quantified.
6. All uncropped raw western data should be uploaded.
7. The title of the manuscript needs to be changed.
8. The authors should check the level of Snail after treating the cells with Fasn inhibitor and also measure colonies and proliferation. What is the lipid level in ACC1KD tumors?
Comments on the Quality of English Language
1. The manuscript needs to be thoroughly edited by a native English-speaking editor.
Author Response
Response to reviewers
Reviewer#2
Comments and Suggestions for Authors
The manuscript titled,” Acetyl-CoA carboxylase 1-mediated protein acetylation and AKT/GSK3β/snail-dependent cell migration in cholangiocarcinoma” has focused on a very interesting topic. My suggestions are as follows:
- The manuscript needs to be edited by a native English-speaking editor.
Authors: Thank you very much for your recommendation. The English language in the manuscript was reviewed and improved by a native English-speaking Prof. James A Will, Emeritus Professor at the University of Wisconsin–Madison, USA.
- For all IP experiment, the IP quality in terms of the antibody used for IP probing should be used for western. Besides this author should use IgG as a control antibody.
Authors: Thank you very much for the shrewd comments. The authors agreed with your recommendation. IgG was included in all immunoprecipitation (IP) experiments to demonstrate the specificity of the protein-antibody interaction. However, to emphasize the IP results of the selected conditions, we have removed “input” and “IgG” control lanes. The original IP results are demonstrated as follows;
All original IP results showing input and IgG control are provided in a separate file with all original Western Blot results.
- Table 1 can be split on the basis of gender.
Authors: Thank you very much for the suggestion. We did not include any table that could be modified based on the reviewer’s recommendation. Could you please provide more specific information?
- Authors should also highlight on the strategies followed internationally and the occurrence trend of these type of infertility
Authors: We are sorry, but our manuscript does not include information about infertility. Would you please provide more details?
- All IHC data should be quantified.
Authors: Thank you very much for the very useful recommendation. The expressions of selected proteins in xenograft tumor tissues were evaluated by immunohistochemistry staining. ACC1, acetylated protein (Ac-K), CK-19, HSP90 and snail are expressed in all cancer cells. ACC1CK-19, HSP90 and snail are localized in the cytoplasm, but Ac-K is observed in both nucleus and cytoplasm. We could only determine the intensity qualitatively as higher or lower expression.
For proliferative marker-Ki-67, it is expressed only in the specific cancer nucleus. So, Ki-67 staining was quantitatively determined by Image J software and presented as %Ki-67-positive nuclei in Supplementary Figure 2 and 3. The related detail was included in the Materials and Methods and Results.
- All uncropped raw western data should be uploaded.
Authors: Thank you very much for the helpful suggestions. All original Western Blot results and original IP results were provided in the separate file.
- The title of the manuscript needs to be changed.
Authors: Thank you very much for the recommendation. The title of the manuscript has been modified to “Augmented global protein acetylation diminishes cell growth and migration of cholangiocarcinoma cells”.
- The authors should check the level of Snail after treating the cells with Fasn inhibitor and also measure colonies and proliferation.
Authors: Thank you very much for your suggestion. The treatment with FASN inhibitor will provide additional information regarding the involvement of de novo fatty acid synthesis and CCA cell properties. However, the relationship between fatty acid synthase (FASN) and cancer cell properties was previously demonstrated in ovarian cancer using FASN-exogenous-overexpressed model (pFASN-transfected cells) and FASN-knockdown model (shFASN-transfected cells) [1]. Ovarian cancer cell growth was measured by colony-formation assay. FASN-overexpressed SW-626 cells acquired higher colony numbers than the parental cells but the colony numbers were lesser in FASN-knockdown SKOV3 cells. Moreover, the involvement of FASN and the expression of mesenchymal markers including N-cadherin, slug, and snail was demonstrated. N- cadherin, slug, and snail were induced in FASN-overexpressed cells, but there were reduced in FASN-knockdown cells.
- What is the lipid level in ACC1KD tumors?
Authors: Thank you very much for your insightful comment. The determination of lipid content in ACC1-KD group is our limitation in the current setting. All tumors were used for formalin-fixed paraffin-embedded (FFPE) tissues. During FFPE preparation, tissues were soaked with paraformaldehyde and xylene for fixation and dehydration purposes, thus they would dissolve lipids in the tissues.
However, the current group previously demonstrated the intracellular neutral lipid content in CCA cells by Oil-Red-O and BODIPY493/503 staining [2]. Reduced neutral lipid contents were observed in ACC1-KD clones, 13ACC1-KD#5 and #18 when compared to KKU-213A parental cells.
Reference
- Jiang, L.; Wang, H.; Li, J.; Fang, X.; Pan, H.; Yuan, X.; Zhang, P., Up-regulated FASN expression promotes transcoelomic metastasis of ovarian cancer cell through epithelial-mesenchymal transition. Int J Mol Sci 2014, 15, 11539-54.
- Saisomboon, S.; Kariya, R.; Boonnate, P.; Sawanyawisuth, K.; Cha'on, U.; Luvira, V.; Chamgramol, Y.; Pairojkul, C.; Seubwai, W.; Silsirivanit, A.; Wongkham, S.; Okada, S.; Jitrapakdee, S.; Vaeteewoottacharn, K., Diminishing acetyl-CoA carboxylase 1 attenuates CCA migration via AMPK-NF-kappaB-snail axis. Biochim Biophys Acta Mol Basis Dis 2023, 1869, 166694.
Comments on the Quality of English Language
- The manuscript needs to be thoroughly edited by a native English-speaking editor.
Authors: Thank you very much for the helpful recommendation. The English language in the current manuscript has been thoroughly reviewed and improved.
Round 2
Reviewer 2 Report
Comments and Suggestions for Authors
Authors have clarified my concerns